# The burden and correlates of bloodborne pathogens in a Zambian Blood Donor Cohort

Mwila Mapipo[1], Alex Maleti[1], Phinnoty Mwansa[2], Paul Daka[1], David Chisompola[3]*

**1** Pathology Laboratory Department, Chinsali General Hospital, Chinsali, Zambia, **2** Pathology Laboratory Department, Unilabs, Lusaka, Zambia, **3** Pathology Laboratory Department, Arthur Davison Children's Hospital, Ndola, Zambia

☯ These authors contributed equally to this work.
* d.chisompola@gmail.com

## Abstract

### Introduction

Transfusion is a critical life-saving intervention, but the safety of the blood supply remains a major public health concern, particularly in sub-Saharan Africa, due to the risk of transfusion-transmitted infections (TTIs). This study aimed to determine the burden and correlates of bloodborne pathogens among blood donors in Muchinga Province, Zambia.

### Methods

A cross-sectional study was conducted using records from 2,667 blood donors at the Chinsali General Hospital Blood Bank between January and December 2021. Data on demographic characteristics, donation type, and ABO/Rhesus blood group were collected. Screening for Hepatitis B (HBV), Hepatitis C (HCV), HIV, and Syphilis was performed using the Abbott Alinity i platform. Logistic regression was used to identify factors associated with infection status.

### Results

The overall prevalence of TTIs among donors was 25.7% (686/2,667). The most prevalent single infections were HBV and Syphilis 8.0% (213/2,667), followed by HCV 7.8% (207/2,667), and HIV 7.7% (206/2,667), with notable co-infection rates, particularly HIV/Syphilis (4.9%). In the composite multivariable analysis, donors with blood group O had 36% lower adjusted odds of any TTI compared to those with blood group A (AOR = 0.64; 95% CI: 0.52–0.79; p < 0.0001) and individuals aged over 45 years had 45% higher adjusted odds of TTIs compared to those aged 16–24 years (AOR = 1.45; 95% CI: 1.04–2.00; p = 0.026). Furthermore, multivariable pathogen-specific analyses revealed distinct risk profiles: donors aged ≥45 years had significantly higher odds of syphilis (AOR = 2.38; 95% CI: 1.49–3.80; p < 0.0001),

which permits unrestricted use, distribution, and reproduction in any medium, provided the original author and source are credited.

**Data availability statement:** All relevant data are within the paper and its Supporting Information files (S2).

**Funding:** The author(s) received no specific funding for this work.

**Competing interests:** The authors have read the journal's policy and have the following competing interests: Author [P.M] is a salaried employee of Unilabs. This does not alter our adherence to the journal's policies on sharing data and materials. The other authors have declared that no competing interests exist.

female sex was associated with lower odds of HBV (AOR = 0.69; 95% CI: 0.52–0.93; p = 0.016), blood group O was protective against HCV (AOR = 0.49; 95% CI: 0.35–0.69; p < 0.0001), and donors from Mpika district had higher odds of HIV (AOR = 1.82; 95% CI: 1.03–3.21; p = 0.037).

## Conclusion

This study highlights a substantial burden of TTIs among blood donors and identifies blood group O as a potential protective factor, suggesting a potential biological basis that requires further study. Strengthened donor screening and public health interventions are critical to improving blood safety.

## Introduction

Blood transfusion remains a critical, life-saving medical intervention that is essential in the management of various clinical conditions, including perioperative and traumatic hemorrhage, anemia, and obstetric complications [1]. However, the safety of blood supply continues to be a significant global public health concern, particularly due to the risk of transfusion-transmitted infections (TTIs) [2]. The four major TTIs of global concern include human immunodeficiency virus (HIV), hepatitis B virus (HBV), hepatitis C virus (HCV), and syphilis, which can be transmitted through contaminated blood products if adequate screening measures are not implemented [3].

The World Health Organization (WHO) emphasizes that ensuring the safety and adequacy of blood supply requires comprehensive strategies including appropriate donor selection, mandatory screening for TTIs, and implementation of quality management systems [4]. Despite significant advances in blood screening technologies and donor selection criteria, TTIs remain a persistent challenge, particularly in resource-limited settings where the burden of these infections in the general population is high [5]. The prevalence of TTIs among blood donors serves as an important epidemiological indicator, reflecting not only the safety of the blood supply but also the underlying disease burden in the donor population and the broader community [6].

Sub-Saharan Africa bears a disproportionate burden of TTIs, with pooled prevalence estimates indicating hepatitis B at 3.0%, HIV at 2.0%, hepatitis C at 1.0%, and syphilis at 2.0% among blood donors in the Southern African Development Community (SADC) region [4]. These rates are substantially higher than those observed in high-income countries, a disparity driven by factors including higher background infection rates in the community, donor recruitment strategies that rely more heavily on higher-risk replacement and first-time donors, and persistent gaps in the implementation and sustainability of screening systems [7].

In Zambia, like many other sub-Saharan African countries, blood safety faces multiple challenges that contribute to high discard rates of donated blood units. The country's blood transfusion services operate within a context of limited resources, where the demand for safe blood products often exceeds supply [8]. The high prevalence of TTIs in the general population, combined with the continued reliance on

replacement donors and family members rather than regular voluntary non-remunerated donors, contributes to elevated TTI rates among blood donors [9]. Operational challenges within the blood safety system, including challenges with limitations in screening infrastructure, intermittent reagent supply, equipment maintenance issues, and constraints in quality assurance systems, further strain the ability to maintain a consistent and safe blood supply, potentially leading to higher rates of TTI-positive donations entering the donor pool [10].

The burden of TTIs in blood donors has significant implications beyond individual patient safety. High discard rates due to TTI-positive donations reduce the available blood supply, potentially compromising the ability to meet clinical needs [11]. Understanding the factors contributing to TTI prevalence and blood unit discards is therefore crucial for developing targeted interventions to improve both blood safety and supply adequacy [12].

This study aims to determine the burden and correlates of transfusion-transmissible infections among blood donors in a Zambian cohort, with particular focus on identifying demographic and donor-related factors associated with TTI prevalence. By elucidating these factors, the study seeks to inform strategies for improving blood safety and reducing TTI-related discards in the region.

## Materials and methods

### Study design and setting

A cross-sectional retrospective study was conducted utilizing blood donor records at Chinsali General Hospital-Muchinga Province Blood Bank center between January 2021 and December 2021. Data from January to December 2021 were extracted in a one-time sampling event between August 1 and September 30, 2023. This study included major blood collection centers in the districts of Isoka, Chinsali, Mafinga, and Mpika. The study was conducted and reported in accordance with the Strengthening the Reporting of Observational Studies in Epidemiology (STROBE) guidelines (S1 Checklist) [13].

### Study population and sampling

The study population consisted of all individuals who presented for blood donation at the participating centers during the study period. From the total of 3058 participants, 2,667 blood donors were included in the final analysis, while 391 donors with incomplete or missing key demographic or clinical data were excluded.

### Data collection and variables

Data were obtained from routine blood donor records and screening registers. It is important to note that these registers were designed for operational tracking and not for comprehensive socio-demographic analysis. While fields for age, sex, donation type, and blood group were consistently completed, the fields for 'education level' and 'marital status' were non-standardized and inconsistently recorded, resulting in a high proportion of missing data ('not documented'). Notably, the data for education status was almost entirely missing, which resulted in a single uniform category (e.g., secondary school) for the few entries that existed; therefore, this variable was omitted from multivariable analysis conducted. These variables were not derived from a structured donor health assessment questionnaire. Personal identifiers like names were not retrieved from the registers in order to ensure anonymity of the donors. The primary outcome variable was infection screening status, defined as a composite measure of infected blood, indicating a positive screening result for one or more transfusion-transmissible infections (TTIs). These TTIs included Hepatitis B Virus (HBV), Hepatitis C Virus (HCV), Human Immunodeficiency Virus (HIV), and Syphilis.

### Serological analysis

Screening for transfusion-transmissible infections was performed using the Abbott Alinity i platform (Abbott Park, Illinois, USA) and the following specific chemiluminescent microparticle immunoassays (CMIA):

- Hepatitis B surface antigen (HBsAg) (Abbott CMIA) for Hepatitis B Virus (HBV)

- Anti-HCV (Abbott CMIA) for Hepatitis C Virus (HCV)

- Anti-HIV Ag/Ab Combo (Abbott CMIA) for Human Immunodeficiency Virus (HIV)

- Syphilis TP (Abbott CMIA) for Treponema pallidum antibodies

   All testing was conducted in accordance with the standard operating procedures of the Zambia National Blood Transfusion Service at Chinsali General Hospital and followed the manufacturer's instructions, including the use of specified calibrators and quality controls to ensure assay validity.

## Statistical analysis

Before analysis, data were cleaned and validated using Microsoft Excel (version 2019) to identify and correct inconsistencies, missing values, and entry errors prior to import into STATA for statistical analysis. Data were analyzed using STATA version 15. Descriptive statistics were used to summarize participant characteristics. Categorical variables were presented as frequencies and percentages, and continuous variables (like Age) as medians and interquartile ranges (IQR) due to non-normal distribution. The association between each independent and the outcome was first assessed using Univariable analyses: Chi-square tests or fishers exact test for small cel counts was used for categorical variables and Mann-Whitney U test for continuous variables. All variables' were included in the multivariable logistic regression model to identify independent predictors. The model was adjusted for potential confounders, and the results were presented as Odds Ratios (OR) and Adjusted Odds Ratios (AOR) with their 95% Confidence Intervals (CI). A p-value of less than 0.05 was considered statistically significant.

## Ethical considerations

Ethical approval for this study was obtained from the University of Zambia School of Medicine Research Ethics Committee on 12th July 2023 (Reference No. 193-07-2023). The study utilized secondary data extracted from existing medical records; hence, no personally identifiable information was collected or recorded in the data collection forms. Given that the study involved a retrospective analysis of anonymized data, the requirement for written or verbal informed consent was waived by the ethics committee. All study procedures were conducted in accordance with the ethical principles outlined in the Declaration of Helsinki

## Results

### Participants characteristics

A total of 2,667 participants were included in the analysis, of whom 686 (25.7%) screened reactive for at least one transfusion-transmissible infection. The baseline characteristics of the study participants are summarized in Table 1. The study population was distributed across three age categories. Individuals aged 16–24 years formed the largest cohort, accounting for 1,401 (52.5%) of all participants. Among them, the infection prevalence was 25.1%. The 25–44 years age group comprised 1,053 individuals (39.5%), exhibiting a nearly identical infection rate of 25.3%. In contrast, the smallest age group, those over 45 years old, with 213 (8.0%) participants, demonstrated the highest observed infection rate at 31.9%. Age group distributions did not differ significantly by infection status (p = 0.097). Overall, the cohort exhibited a nearly equal sex distribution, with males accounting for 53.2% of the study population. Univariable analysis revealed only one factor that was significantly associated with blood infection status: blood group. Furthermore, the prevalence of infection varied significantly by blood group (p < 0.0001). Blood groups A (29.2%) and B (32.0%) demonstrated the highest prevalence, while blood group O had the lowest (21.9%). No statistically significant associations were found with age, sex, marital status, district, donor type (first-time vs. repeat), or Rhesus status.

**Table 1. Participant characteristics.**

| Variable | Median (IQR) / Frequency (%) | Infected blood | | p-value |
|---|---|---|---|---|
| | | Yes = 686 (25.7) | No = 1,981 (74.3) | |
| **Age group, Years** | | | | |
| *16-24 years* | 1,401 (52.5) | 352 (25.1) | 1,049 (74.9) | 0.097 |
| *25-44 years* | 1,053 (39.5) | 266 (25.3) | 787 (74.7) | |
| *> 45 years* | 213 (8.0) | 68 (31.9) | 145 (68.1) | |
| **Sex** | | | | |
| *Male* | 1,419 (53.2) | 379 (26.7) | 1,040 (73.3) | 0.214 |
| *Female* | 1,248 (46.8) | 307 (24.6) | 941 (75.4) | |
| **Education** | | | | |
| *Secondary* | 314 (11.8) | 99 (31.5) | 215 (68.5) | – |
| **Marital status** | | | | |
| *Married* | 4 (0.2) | 1 (25.0) | 3 (75.0) | 0.135 |
| *Not Married* | 302 (11.3) | 92 (30.5) | 210 (69.5) | |
| **District** | | | | |
| *Isoka* | 329 (12.3) | 86 (26.1) | 243 (73.9) | 0.184 |
| *Chinsali* | 95 (3.6) | 16 (16.8) | 79 (83.2) | |
| *Mafinga* | 40 (1.5) | 8 (20.0) | 32 (80.0) | |
| *Mpika* | 2,203 (82.6) | 576 (26.1) | 1,627 (73.9) | |
| **Type of donation** | | | | |
| *First time* | 632 (23.7) | 158 (25.0) | 474 (75.0) | 0.0635 |
| *Repeat* | 2,035 (76.3) | 528 (26.0) | 1,507 (74.0) | |
| **Rhesus** | | | | |
| *Positive* | 2,657 (99.6) | 682 (25.7) | 1,975 (74.3) | 0.301 |
| *Negative* | 10 (0.4) | 4 (40.0) | 6 (60.0) | |
| **Blood group** | | | | |
| *A* | 664 (24.9) | 194 (29.2) | 470 (70.8) | **<0.0001** |
| *B* | 490 (18.3) | 157 (32.0) | 333 (68.0) | |
| *AB* | 82 (3.1) | 21 (25.6) | 61 (74.4) | |
| *O* | 1,431 (53.7) | 314 (21.9) | 1,117 (78.1) | |

### Factors associated with infected blood screening results

To identify independent predictors of blood infection, an univariable and multivariable logistic regression was performed, the results of which are presented in Table 2. At univariable analysis, age > 45 years (OR: 1.40, 95% CI: 1.02–1.91, p = 0.036), blood group O (OR: 0.68, 95% CI: 0.55–0.83, p < 0.0001) were significantly associated with the outcome. Multi-variable model included all variables; age > 45 years (AOR: 1.45, 95% CI: 1.04–2.00, p = 0.026), and blood group O (AOR: 0.64, 95% CI: 0.52–0.79, p < 0.0001) remained statistically significant after adjustment for all other variables, including sex, marital status, district, donor type, Rhesus factor, and other blood groups (Table 2).

### Prevalence of blood screened by infection detected

Among the 2,667 donors analyzed, 686 (25.7%) tested reactive for at least one TTI. As detailed in Table 2, the most prevalent infections were HBV and syphilis, each with 213 cases and a prevalence of 8.0%. These were followed by HCV (207 cases; 7.8%) and HIV (206 cases; 7.7%) (Table 3).Co-infections with two pathogens were also observed, the most

**Table 2.  Logistic regression for Factors Associated with Infected Blood Screening Results.**

| Variable | OR (95%CI) | P value | AOR (95%CI) | P value |
|---|---|---|---|---|
| **Age group, Years** | | | | |
| *16-24 years* | Ref | | Ref | |
| *25-44 years* | 1.01 (0.83, 1.21) | 0.939 | 1.06 (0.86, 1.29) | 0.576 |
| *> 45 years* | 1.40 (1.02, 1.91) | **0.036** | 1.45 (1.04, 2.00) | **0.026** |
| **Sex** | | | | |
| *Male* | Ref | | Ref | |
| *Female* | 0.89 (0.75, 1.06) | 0.214 | 0.94 (0.79, 1.13) | 0.539 |
| **Marital status** | | | | |
| *Married* | Ref | | Ref | |
| *Not Married* | 1.31 (0.13, 12.8) | 0.814 | 0.33 (0.02, 4.31) | 0.396 |
| **District** | | | | |
| *Isoka* | Ref | | Ref | |
| *Chinsali* | 0.57 (0.32, 1.03) | 0.064 | 0.61 (0.33, 1.11) | 0.108 |
| *Mafinga* | 0.70 (0.31, 1.59) | 0.402 | 0.72 (0.32, 1.65) | 0.448 |
| *Mpika* | 1.00 (0.76, 1.30) | 0.998 | 0.64 (0.51, 0.79) | 0.706 |
| **Type of donation** | | | | |
| *First time* | Ref | | Ref | |
| *Repeat* | 1.05 (0.85, 1.29) | 0.635 | 1.13 (0.90, 1.43) | 0.281 |
| **Rhesus** | | | | |
| *Positive* | Ref | | Ref | |
| *Negative* | 0.51 (0.14, 1.84) | 0.309 | 0.49 (0.13, 1.76) | 0.271 |
| **Blood group** | | | | |
| *A* | Ref | – | Ref | – |
| *B* | 1.14 (0.88, 1.47) | 0.303 | 1.06 (0.82, 1.38) | 0.624 |
| *AB* | 0.83 (0.49, 1.40) | 0.497 | 0.78 (0.46, 1.33) | 0.371 |
| *O* | 0.68 (0.55, 0.83) | <0.0001 | 0.64 (0.52, 0.79) | **<0.0001** |

Marital status was poorly documented, with data available for only 306 participants (11.5%). The 'not documented' category (n = 2,361) was excluded from regression analyses to prevent misclassification.

common being HIV/Syphilis (4.9%, n = 34), HBV/Syphilis (3.5%, n = 24), and HCV/Syphilis (2.9%, n = 20). Complex co-infections involving three or four pathogens were rare, each constituting less than 1% of the infected cohort.

## Pathogen-Specific Risk Factor Analyses

Separate multivariable logistic regression analyses were conducted for each TTI to identify pathogen-specific risk factors. The results of these adjusted analyses are presented in Table 4. Age over 45 years emerged as a strong, independent risk factor for syphilis, with donors in this age group having more than twice the odds of a reactive syphilis test compared to donors aged 16–24 (AOR: 2.38, 95% CI: 1.49–3.80, p < 0.0001). This age association was not observed for HIV, HBV, or HCV. Female sex was associated with significantly lower odds of HBV reactivity (AOR: 0.69, 95% CI: 0.52–0.93, p = 0.016) (Table 4). A similar protective trend was observed for HCV, though it did not reach statistical significance. Notable geographic variation was observed for HIV, with donors from Mpika district showing significantly higher adjusted odds of HIV reactivity (AOR: 1.82, 95% CI: 1.03–3.21, p = 0.037). Blood group O was associated with substantially reduced odds of HCV reactivity (AOR: 0.49, 95% CI: 0.35–0.69, p < 0.0001). While a protective trend was observed for HBV (AOR: 0.77,

**Table 3. Distribution of Reactive Results by Pathogen.**

| Pathogens | Frequency (n) | Percentage (%) |
|---|---|---|
| HBV | 213 | 8.0 |
| Syphilis | 213 | 8.0 |
| HCV | 207 | 7.8 |
| HIV | 206 | 7.7 |
| HBV/HCV | 17 | 0.6 |
| HIV/HBV | 19 | 0.7 |
| HEPB/Syphilis | 24 | 0.9 |
| HIV/HVC | 15 | 0.6 |
| HIV/Syphilis | 34 | 1.3 |
| HCV/Syphilis | 20 | 0.7 |
| Syphilis/HBV/HCV | 5 | 0.2 |
| HIV/HCV/Syphilis | 6 | 0.2 |
| HIV/HCV/HBV | 2 | 0.1 |
| HIV/HCV/HBV/syphilis | 2 | 0.1 |

Abbreviations: HBV, Hepatitis B; HCV, Hepatitis C, and HIV: Human Immunodeficiency virus

p = 0.157), it did not reach statistical significance in this analysis. Marital status (not married) showed significant protective associations for syphilis and HCV. Donor type (repeat vs. first-time) and Rhesus factor were not significantly associated with any pathogen.

## Discussion

This study provides a detailed analysis of the prevalence and correlates of transfusion-transmissible infections among blood donors in Muchinga Province of Zambia. The key findings reveal a high overall infection prevalence of 25.7%, with significant independent associations between infection status and ABO blood group.

The overall prevalence of transfusion-transmissible infections (TTIs) in this study was 25.7%. This rate is consistent with findings from Burkina Faso [14], suggesting a similarly high burden of TTIs in certain West and Central African regions. However, our prevalence is substantially higher than rates reported in studies from Uganda, Ethiopia, Nigeria and India [15–18]. This disparity underscores a significant and geographically variable threat to blood safety within sub-Saharan Africa. The observed variations are likely attributable to differences in regional disease epidemiology, donor population characteristics (e.g., the proportion of first-time versus repeat donors), and the sensitivity of serological testing assays employed.

Interestingly, our observed findings of the independent protective role of blood group O. Donors with blood group O had 36% lower adjusted odds of being infected compared to those with blood group A (AOR = 0.64, 95% CI: 0.52–0.79). This finding is consistent with a growing body of evidence suggesting a biological link between the ABO blood group system and susceptibility to certain infections [19–22]. The absence of certain glycan antigens may reduce the ability of pathogens to attach and enter host cells or may modulate the host's inflammatory and immune response to infection [23]. For instance, previous studies have linked blood group O to a reduced risk of severe malaria [24] and SARS-CoV-2 infection [25], while non-O blood groups have a higher risk SARS-CoV-2 infections [26]. Our results extend this concept to a broad spectrum of TTIs, including HBV, HCV, HIV, and Syphilis, within a donor population. Although the observed association suggests a possible protective biological mechanism, causal inference cannot be established due to the cross-sectional nature of the data. This warrants further investigation into the specific mechanisms that might confer this protective advantage.

**Table 4. Pathogen-Specific Risk Factor Analyses.**

| Variable | HIV | | Syphilis | | Hepatitis B | | Hepatitis C | |
|---|---|---|---|---|---|---|---|---|
| | AOR (95%CI) | P value | AOR (95%CI) | P value | AOR (95%CI) | P value | AOR (95%CI) | P value |
| **Age group, Years** | | | | | | | | |
| *16-24 years* | Ref | | Ref | | Ref | | Ref | |
| *25-44 years* | 1.15 (0.81, 1.61) | 0.415 | 1.26 (0.90, 1.75) | 0.168 | 0.84 (0.61, 1.17) | 0.318 | 1.32 (0.96, 1.82) | 0.080 |
| *> 45 years* | 1.22 (0.70, 2.13) | 0.478 | 2.38 (1.49, 3.80) | **<0.0001** | 1.03 (0.61, 1.75) | 0.888 | 1.35 (0.79, 2.28) | 0.261 |
| **Sex** | | | | | | | | |
| *Male* | Ref | | Ref | | Ref | | Ref | |
| *Female* | 1.12 (0.84, 1.51) | 0.426 | 1.17 (0.87, 1.56) | 0.276 | 0.69 (0.52, 0.93) | **0.016** | 0.79 (0.58, 1.05) | 0.116 |
| **Marital status** | | | | | | | | |
| *Married* | Ref | | Ref | | Ref | | Ref | |
| *Not Married* | 0.44 (0.12, 1.61) | 0.218 | 0.14 (0.04, 0.49) | **0.002** | 0.28 (0.01, 6.39) | 0.429 | 0.18 (0.04, 0.80) | **0.024** |
| **District** | | | | | | | | |
| *Isoka* | Ref | | Ref | | Ref | | Ref | |
| *Chinsali* | 0.24 (0.03, 1.89) | 0.178 | 0.81 (0.34, 1.94) | 0.649 | 0.63 (0.25, 1.59) | 0.337 | 0.81 (0.30, 2.20) | 0.686 |
| *Mafinga* | 0.57 (0.07, 4.51) | 0.597 | 0.82 (0.23, 2.83) | 0.754 | – | – | 1.52 (0.49, 4.66) | 0.463 |
| *Mpika* | 1.82 (1.03, 3.21) | **0.037** | 0.73 (0.48, 1.11) | 0.150 | 0.73 (0.48, 1.10) | 0.140 | 1.17 (0.74, 1.83) | 0.487 |
| **Type of donation** | | | | | | | | |
| *First time* | Ref | | Ref | | Ref | | Ref | |
| *Repeat* | 1.25 (0.84, 1.85) | 0.253 | 1.39 (0.93, 2.07) | 0.103 | 1.20 (0.82, 1.74) | 0.335 | 0.78 (0.54, 1.12) | 0.181 |
| **Rhesus** | | | | | | | | |
| *Positive* | Ref | | Ref | | Ref | | Ref | |
| *Negative* | 0.27 (0.05, 1.35) | 0.112 | 0.79 (0.09, 6.36) | 0.826 | 0.75 (0.09, 6.16) | 0.796 | 0.69 (0.08, 5.62) | 0.732 |
| **Blood group** | | | | | | | | |
| *A* | Ref | – | Ref | – | Ref | – | Ref | – |
| *B* | 1.36 (0.89, 2.08) | 0.152 | 1.45 (0.95, 2.22) | 0.077 | 1.01 (0.66, 1.54) | 0.945 | 0.74 (0.49, 1.11) | 0.686 |
| *AB* | 1.3 (0.44, 2.39) | 0.940 | 0.83 (0.31, 2.16) | 0.703 | 0.74 (0.31, 1.80) | 0.519 | 0.74 (0.33, 1.69) | 0.486 |
| *O* | 0.79 (0.54, 1.16) | 0.246 | 0.97 (0.67, 1.39) | 0.881 | 0.77 (0.54, 1.10) | 0.157 | 0.49 (0.35, 0.69) | **<0.0001** |

Note: Dashes (-) indicate categories with no reactive cases, preventing model estimation. Bold text highlights statistically significant associations (p < 0.05). Models adjusted for all variables listed.

AOR: Adjusted Odds Ratio; CI: Confidence Interval.

The profile of infections, dominated by Hepatitis B and Syphilis (8%), followed by Hepatitis C (7.8%), and HIV (7.7%), paints a picture of a population burdened by multiple endemics sexually transmitted and blood-borne infections. The high rate of co-infections, particularly HIV/Syphilis (4.9%), further complicates the clinical picture and suggests intertwined transmission networks. This prevalence is considerably higher than reports from high-income countries and aligns with studies from other parts of sub-Saharan Africa [14,18], reflecting regional health disparities and the urgent need for strengthened public health interventions, including vaccination programs (e.g., HBV), safe injection practices, and sexual health education (5).

Our pathogen-specific analyses revealed both distinct and shared epidemiological patterns among the TTIs. The strong association between older age (>45 years) and syphilis reactivity aligns with trends observed in other settings such as sub-Saharan African and Asia, possibly reflecting cumulative lifetime exposure risk or age-cohort effects in sexual behavior [27,28]. Conversely, female sex demonstrated a specific protective effect against HBV, which may be linked to differential exposure to traditional risk factors such as percutaneous injuries or historical vaccination patterns in Zambia.

This finding warrants careful interpretation and highlights the complexity of using routine administrative data for risk characterization.

The biological correlate of blood group O emerged as a significant protective factor specifically for Hepatitis C, corroborating emerging evidence of an association between ABO phenotypes and susceptibility to specific viral infections [29]. This may suggest a role for natural antibodies or other glycobiology mechanisms in HCV pathogenesis or clearance. Geographically, donors from Mpika district had significantly higher adjusted odds of HIV reactivity compared to Isoka. This district-level heterogeneity underscores the importance of localized HIV epidemiology and may reflect differences in local prevalence, testing accessibility, or the demographic composition of the donor base across collection centers. Importantly, donor type (repeat vs. first-time) was not a significant predictor for any pathogen in the adjusted models, suggesting that in this population, the traditional marker of a "safer" repeat donor may not hold once key demographic and geographic factors are accounted for. This challenges a common assumption in blood safety strategies and indicates that donor selection criteria may need to be context-specific.

### Limitations

This study had several limitations. Firstly, its cross-sectional design allows for the identification of associations but not causal relationships. Additionally, this study did not examine non-TTI reasons for blood discard (such as expiry, inadequate volume, or storage issues), which also contribute to blood unit wastage. Future studies should consider a broader assessment of all factors leading to blood discards to fully inform blood safety and supply policies. Secondly, although we adjusted for available demographic and donation-related variables, residual and unmeasured confounding remains possible. Important behavioral, socioeconomic, and clinical factors, such as sexual behavior, injection drug use, occupation, access to healthcare, vaccination history (e.g., for HBV), and prior diagnosis or treatment for TTIs, were not available in the donor records and could influence both exposure and outcome. Thirdly, as noted, the high proportion of "not documented" for variables like education and marital status introduces potential misclassification and residual confounding. Fourthly, the use of a composite "infected blood" outcome, while useful for a broad safety screen, groups together pathogens with different transmission dynamics. Fifthly, while we conducted comprehensive analyses of all available demographic and donation-related variables, we were unable to analyze quantitative serological titer data as these were not routinely recorded in our donor screening registries. Future studies incorporating titer levels could provide additional insights into infection intensity and recency. A key limitation of this study is the incomplete documentation of socioeconomic variables, particularly education level and marital status. Education data were largely missing, with secondary education being the only recorded category for a minority of participants, while marital status was similarly poorly documented. Future studies should ensure comprehensive and standardized collection of socioeconomic data to enable more robust and interpretable analyses. Additionally, the study was conducted in Muchinga Province of Zambia, which may limit the generalizability of the findings to the entire country or other settings. Finally, potential selection bias cannot be ruled out, as blood donors may not be representative of the general population due to eligibility screening and self-selection factors.

### Conclusion

This study reveals a high burden of transfusion-transmissible infections among blood donors in Zambia. It identifies blood group O as a significant independent protective factor, a finding with potential biological implications that merit further research. The high prevalence rate signals an urgent need to reinforce national blood safety strategies, including rigorous donor screening, hemovigilance, and targeted public health measures to reduce the community burden of these infections. Future research should focus on prospective designs with more comprehensive data collection to better understand the sociodemographic and biological determinants of infection risk in African blood donor populations.

## Supporting information

**S1 Checklist. Strobe checklist.**
(DOCX)

**S2 Data. Dataset.**
(XLSX)

## Author contributions

**Conceptualization:** Mwila Mapipo, Alex Maleti.

**Data curation:** Mwila Mapipo, Alex Maleti, Phinnoty Mwansa, Paul Daka, David Chisompola.

**Formal analysis:** Phinnoty Mwansa, David Chisompola.

**Investigation:** Mwila Mapipo, Alex Maleti, Phinnoty Mwansa, David Chisompola.

**Methodology:** Mwila Mapipo, Alex Maleti.

**Resources:** Mwila Mapipo.

**Supervision:** David Chisompola.

**Validation:** Paul Daka.

**Visualization:** Mwila Mapipo, Alex Maleti, David Chisompola.

**Writing – original draft:** Mwila Mapipo, Alex Maleti, Phinnoty Mwansa, David Chisompola.

**Writing – review & editing:** Mwila Mapipo, Alex Maleti, Phinnoty Mwansa, Paul Daka, David Chisompola.

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
