## [Decision Letter · Decision Letter 0]

23 Dec 2025

PONE-D-25-55464The Burden and Correlates of Bloodborne Pathogens in a Zambian Blood Donor CohortPLOS One

Dear Dr. Chisompola,

Thank you for submitting your manuscript to PLOS ONE. After careful consideration, we feel that it has merit but does not fully meet PLOS ONE’s publication criteria as it currently stands. Therefore, we invite you to submit a revised version of the manuscript that addresses the points raised during the review process.

We look forward to receiving your revised manuscript.

Kind regards,

Nneoma Confidence JeanStephanie Anyanwu, Ph.D.

Academic Editor

PLOS One

Journal Requirements:

[The authors have declared that no competing interests exist.].

We note that one or more of the authors are employed by a commercial company: Unilabs.

Additional Editor Comments:

In addition to addressing the reviewers' comments, I strongly suggest analysing each factor and the titres of serological tests for each pathogen. An analysis limited to the educational status and blood grouping will bias the results.

Reviewers' comments:

Reviewer's Responses to Questions

**Comments to the Author**

1. Is the manuscript technically sound, and do the data support the conclusions?

Reviewer #1: Partly

Reviewer #2: Partly

2. Has the statistical analysis been performed appropriately and rigorously? 

Reviewer #1: I Don't Know

Reviewer #2: Yes

3. Have the authors made all data underlying the findings in their manuscript fully available?

The PLOS Data policy requires authors to make all data underlying the findings described in their manuscript fully available without restriction, with rare exception (please refer to the Data Availability Statement in the manuscript PDF file). The data should be provided as part of the manuscript or its supporting information, or deposited to a public repository. For example, in addition to summary statistics, the data points behind means, medians and variance measures should be available. If there are restrictions on publicly sharing data—e.g. participant privacy or use of data from a third party—those must be specified.requires authors to make all data underlying the findings described in their manuscript fully available without restriction, with rare exception (please refer to the Data Availability Statement in the manuscript PDF file). The data should be provided as part of the manuscript or its supporting information, or deposited to a public repository. For example, in addition to summary statistics, the data points behind means, medians and variance measures should be available. If there are restrictions on publicly sharing data—e.g. participant privacy or use of data from a third party—those must be specified.requires authors to make all data underlying the findings described in their manuscript fully available without restriction, with rare exception (please refer to the Data Availability Statement in the manuscript PDF file). The data should be provided as part of the manuscript or its supporting information, or deposited to a public repository. For example, in addition to summary statistics, the data points behind means, medians and variance measures should be available. If there are restrictions on publicly sharing data—e.g. participant privacy or use of data from a third party—those must be specified.requires authors to make all data underlying the findings described in their manuscript fully available without restriction, with rare exception (please refer to the Data Availability Statement in the manuscript PDF file). The data should be provided as part of the manuscript or its supporting information, or deposited to a public repository. For example, in addition to summary statistics, the data points behind means, medians and variance measures should be available. If there are restrictions on publicly sharing data—e.g. participant privacy or use of data from a third party—those must be specified.

Reviewer #1: No

Reviewer #2: No

4. Is the manuscript presented in an intelligible fashion and written in standard English?

Reviewer #1: Yes

Reviewer #2: Yes

5. Review Comments to the Author

Reviewer #1: The results of this study conducted in Zambia were as expected as recent high-quality studies and reports (systematic reviews, regional analyses, and large reports) from several African countries that identify the burden of bloodborne pathogens (particularly hepatitis B virus, hepatitis C virus, HIV, and syphilis/other bloodborne diseases like malaria) and examine their associations. Also, recent studies from Africa and other regions have examined whether ABO blood type (particularly blood type O) is associated with lower TTI risk. Multiple studies (from Sub-Saharan Africa, parts of Asia, and other low- and middle-income countries) find that lower education (below secondary) is associated with a higher prevalence of blood-borne infections among blood donors, while donors with secondary or higher education generally show lower infection rates — but there are important regional and methodological exceptions.

What's striking is that limitations of this study make it extremely is not reliable.

The study method was a cross-sectional study conducted using records, Data were obtained from routine blood donor records and screening registers. In regards to the level of education as a risk factor of transfusion-transmissible infections (TTIs), the author did not clarify whether there were using a structured/ standard blood transfusion service questionnaire for Socio-demographic characteristics and associated risk factors (educational level). If such a questionnaire is used in the study, it will be shown numbers of donors with primary and higher education and not just secondary. It is well known that education level is not a requirement for routine blood donation. Blood donation screening records refer to the process of using a register of previously deferred donors to check potential donors and prevent ineligible individuals from donating blood. Yes, the donor's educational level can be a factor in blood donation screening in some studies (a questionnaire is conducted beforehand for this purpose), although it is not used in the basic health screening process itself. The primary purpose of the screening is to ensure blood safety by assessing the donor's health. This is done through questionnaires and a confidential interview that checks their medical history, lifestyle, and physical health measures such as blood pressure, pulse, and hemoglobin levels.

Here few comments:

In Table 1. The median age of the cohort was 24 years, is this median age for infected and non infected (not mention in table), Could authors add the frequency of additional ages and corelate them to the level of education (secondary)?

0

Reviewer #2: This manuscript investigates the predictors of the infected blood donors with four pathogens. In fact, the authors analysed only two factors; the educational status and the blood groups. The authors missed to analyse many factors that listed in the table 2. In addition, the presented data are categorized rather than continuous which described rather than to analyzes the results Accordingly, the presented results will not free from bias.

Using appropriate statistical analysis will improve the quality of the study

The titles of the serological tests need to be added and analyzed

The figures are not informative, and needs more clarifications and improvements.

6. PLOS authors have the option to publish the peer review history of their article (what does this mean?). If published, this will include your full peer review and any attached files.). If published, this will include your full peer review and any attached files.). If published, this will include your full peer review and any attached files.). If published, this will include your full peer review and any attached files.

...

Reviewer #1: No

Reviewer #2: No

---

## [Author Response · Author response to Decision Letter 1]

26 Dec 2025

PONE-D-25-55464: The Burden and Correlates of Bloodborne Pathogens in a Zambian Blood Donor Cohort

Journal Requirements:

Response: We thank the editorial team for this comment. The manuscript has been formatted to the journal style requirements.

Response: We thank the editorial team for this comment. The data availability statement has been revised. The raw data underlying the results presented in the study have been uploaded as a Supporting information file (S2).

[The authors have declared that no competing interests exist.].

We note that one or more of the authors are employed by a commercial company: Unilabs.

Response: We thank the editorial team for this comment. The Competing Interests section has been revised. The authors have read the journal's policy and have the following competing interests: Author [P.M] is a salaried employee of Unilabs. This does not alter our adherence to the journal's policies on sharing data and materials. The other authors have declared that no competing interests exist.

Response: We thank the editorial team for this comment. We have revised the funding statement. This research received no specific grant from any funding agency in the public, commercial, or not-for-profit sectors. The authors [P.M] is employed by Unilabs, but Unilabs provided no financial support specifically for this work. Moreover, we have verified the 'Author Contributions' section. The contributions of author [P.M] employed by Unilabs are articulated in terms of their direct scientific roles in this research such as data analysis, and manuscript drafting), confirming their involvement was academic in nature.

Response: We thank the editorial team for this comment. The Competing Interests section has been revised. The authors have read the journal's policy and have the following competing interests: Author [P.M] is a salaried employee of Unilabs. This does not alter our adherence to the journal's policies on sharing data and materials. The other authors have declared that no competing interests exist.

Response: We thank the editorial team for this comment. We have revised the cover letter to include funding statement and competing interests

Response: We thank the editorial team for this comment.

Additional Editor Comments:

In addition to addressing the reviewers' comments, I strongly suggest analysing each factor and the titres of serological tests for each pathogen. An analysis limited to the educational status and blood grouping will bias the results.

Response: We sincerely thank the reviewer for these valuable insights. We have substantially revised our analysis in response to your recommendations:

• We have expanded our multivariable logistic regression model to include all variables in the dataset (age, sex, marital status, education, district, donor type, Rhesus factor, and ABO blood group). The revised analysis no longer focuses exclusively on education and blood group, thus avoiding the potential bias noted by the reviewer.

• We have conducted separate multivariable regression analyses for each specific transfusion-transmissible infection (HBV, HCV, HIV, and Syphilis) to identify whether different factors predict reactivity for different pathogens. These results are presented in a new supplementary table.

• We acknowledge the reviewer's suggestion regarding analysis of serological titers. Unfortunately, our laboratory's routine screening protocol uses qualitative (positive/negative) results for blood donor screening, and quantitative titer data were not systematically recorded in the donor registries. We have clarified this limitation in the manuscript.

Review Comments to the Author

Reviewer #1: The results of this study conducted in Zambia were as expected as recent high-quality studies and reports (systematic reviews, regional analyses, and large reports) from several African countries that identify the burden of bloodborne pathogens (particularly hepatitis B virus, hepatitis C virus, HIV, and syphilis/other bloodborne diseases like malaria) and examine their associations. Also, recent studies from Africa and other regions have examined whether ABO blood type (particularly blood type O) is associated with lower TTI risk. Multiple studies (from Sub-Saharan Africa, parts of Asia, and other low- and middle-income countries) find that lower education (below secondary) is associated with a higher prevalence of blood-borne infections among blood donors, while donors with secondary or higher education generally show lower infection rates — but there are important regional and methodological exceptions.

Response: We thank the reviewer for the kind feedback.

What's striking is that limitations of this study make it extremely is not reliable.

The study method was a cross-sectional study conducted using records, Data were obtained from routine blood donor records and screening registers. In regards to the level of education as a risk factor of transfusion-transmissible infections (TTIs), the author did not clarify whether there were using a structured/ standard blood transfusion service questionnaire for Socio-demographic characteristics and associated risk factors (educational level). If such a questionnaire is used in the study, it will be shown numbers of donors with primary and higher education and not just secondary. It is well known that education level is not a requirement for routine blood donation. Blood donation screening records refer to the process of using a register of previously deferred donors to check potential donors and prevent ineligible individuals from donating blood. Yes, the donor's educational level can be a factor in blood donation screening in some studies (a questionnaire is conducted beforehand for this purpose), although it is not used in the basic health screening process itself. The primary purpose of the screening is to ensure blood safety by assessing the donor's health. This is done through questionnaires and a confidential interview that checks their medical history, lifestyle, and physical health measures such as blood pressure, pulse, and hemoglobin levels.

Response: We thank the reviewer for this important critique on how data was collected. We have revised the methodology section to make this section clearer. It now reads “Data were extracted from routine blood donor screening registers. It is important to note that these registers were designed for operational tracking and not for comprehensive socio-demographic analysis. While fields for age, sex, donation type, and blood group were consistently completed, the fields for 'education level' and 'marital status' were non-standardized and inconsistently recorded, resulting in a high proportion of missing data ('not documented'). These variables were not derived from a structured donor health assessment questionnaire”.

Here few comments:

In Table 1. The median age of the cohort was 24 years, is this median age for infected and non infected (not mention in table), Could authors add the frequency of additional ages and corelate them to the level of education (secondary)?

0

Response: We thank the reviewer for this important suggestion and critique. We have revised table 1 to include age for the infected 24 years (19, 35) and non-infected 23 years (18, 33). We have also added frequency of additional ages. Moreover, while we agree that exploring the relationship between age and education would be informative, education data in our dataset were incompletely documented, with over 88% recorded as ‘not documented’ and only one education category (secondary) available. Performing a cross-tabulation or correlation analysis under these conditions would be statistically unreliable and potentially misleading. We have therefore retained education as a descriptive variable and clarified this limitation in Table 1 footnote, and Discussion.

Reviewer #2: This manuscript investigates the predictors of the infected blood donors with four pathogens. In fact, the authors analysed only two factors; the educational status and the blood groups. The authors missed to analyse many factors that listed in the table 2. In addition, the presented data are categorized rather than continuous which described rather than to analyzes the results Accordingly, the presented results will not free from bias.

Using appropriate statistical analysis will improve the quality of the study

Response: We thank the reviewer for this important suggestion and critique. We have reanalyzed and revised table 2 and have now included all variables from the dataset in a comprehensive multivariable logistic regression model, as presented in the revised Table 2. No predictors were excluded from the analysis. Secondly, the changes made using the full model minimizes unnecessary categorization, directly addressing the potential for omitted variable bias and information loss, strengthening the validity of our findings.

The titles of the serological tests need to be added and analyzed

Response: We thank the reviewer for this important suggestion. We have revised the method section to make the serological tests clearer. We have also revised the figure which was meant to illustrate pathogens analyzed by serological test.

The figures are not informative, and needs more clarifications and improvements.

Response: We thank the reviewer for this important suggestion and critique. We have revised the figure 1 and have converted figure 2 to table 3 to make them clearer and informative.

---

## [Decision Letter · Decision Letter 1]

13 Jan 2026

PONE-D-25-55464R1The Burden and Correlates of Bloodborne Pathogens in a Zambian Blood Donor CohortPLOS One

Dear Dr. Chisompola,

Thank you for submitting your manuscript to PLOS ONE. After careful consideration, we feel that it has merit but does not fully meet PLOS ONE’s publication criteria as it currently stands. Therefore, we invite you to submit a revised version of the manuscript that addresses the points raised during the review process.

We look forward to receiving your revised manuscript.

Kind regards,

Nneoma Confidence JeanStephanie Anyanwu, Ph.D.

Academic Editor

PLOS One

**Journal Requirements:**

Reviewers' comments:

Reviewer's Responses to Questions

**Comments to the Author**

1. If the authors have adequately addressed your comments raised in a previous round of review and you feel that this manuscript is now acceptable for publication, you may indicate that here to bypass the “Comments to the Author” section, enter your conflict of interest statement in the “Confidential to Editor” section, and submit your "Accept" recommendation.

Reviewer #1: (No Response)

Reviewer #3: (No Response)

Reviewer #4: (No Response)

2. Is the manuscript technically sound, and do the data support the conclusions?

Reviewer #1: Partly

Reviewer #3: Partly

Reviewer #4: Partly

3. Has the statistical analysis been performed appropriately and rigorously? 

Reviewer #1: I Don't Know

Reviewer #3: N/A

Reviewer #4: No

4. Have the authors made all data underlying the findings in their manuscript fully available?

The PLOS Data policy requires authors to make all data underlying the findings described in their manuscript fully available without restriction, with rare exception (please refer to the Data Availability Statement in the manuscript PDF file). The data should be provided as part of the manuscript or its supporting information, or deposited to a public repository. For example, in addition to summary statistics, the data points behind means, medians and variance measures should be available. If there are restrictions on publicly sharing data—e.g. participant privacy or use of data from a third party—those must be specified.requires authors to make all data underlying the findings described in their manuscript fully available without restriction, with rare exception (please refer to the Data Availability Statement in the manuscript PDF file). The data should be provided as part of the manuscript or its supporting information, or deposited to a public repository. For example, in addition to summary statistics, the data points behind means, medians and variance measures should be available. If there are restrictions on publicly sharing data—e.g. participant privacy or use of data from a third party—those must be specified.requires authors to make all data underlying the findings described in their manuscript fully available without restriction, with rare exception (please refer to the Data Availability Statement in the manuscript PDF file). The data should be provided as part of the manuscript or its supporting information, or deposited to a public repository. For example, in addition to summary statistics, the data points behind means, medians and variance measures should be available. If there are restrictions on publicly sharing data—e.g. participant privacy or use of data from a third party—those must be specified.requires authors to make all data underlying the findings described in their manuscript fully available without restriction, with rare exception (please refer to the Data Availability Statement in the manuscript PDF file). The data should be provided as part of the manuscript or its supporting information, or deposited to a public repository. For example, in addition to summary statistics, the data points behind means, medians and variance measures should be available. If there are restrictions on publicly sharing data—e.g. participant privacy or use of data from a third party—those must be specified.

Reviewer #1: No

Reviewer #3: No

Reviewer #4: Yes

5. Is the manuscript presented in an intelligible fashion and written in standard English?

Reviewer #1: Yes

Reviewer #3: No

Reviewer #4: No

6. Review Comments to the Author

Reviewer #1: The author to some extent has responded to the comments, It's a decent effort , though it lacks depth. Just here one comment, in tables 1, 2, age group 14-24 years. Does 14 years eligible for blood donation in Zambia?!

Usually Standard age of donation is 18 to 65 years old, as organizations focus on donors who are in good health, meet weight standards, hemoglobin standards, and have a stable medical history to ensure donor safety and blood quality.

The minimum age is usually 16 or 17 with parental consent in some places. But at 14 it is restricted as individuals are still developing, and to protect their health.

Reviewer #3: This cross sectional study present the burden and correlates of bloodborne pathogens among blood donors in a Zambian cohort, particular focus on identifying demographic, behavioral, and operational factors associated with transfusion-transmitted infections (TTI) prevalence. However, my concerns/suggestions are not necessarily a reflection of the quality of the work, as follows:

•Overall, the introduction would benefit from a clearer and more cohesive rationale, with smoother transitions between paragraphs. In particular, the discussion of cancer treatment implications lacks a sufficiently developed biological or clinical rationale. Strengthening the mechanistic or pathophysiological basis for this link would help justify the study’s relevance and potential impact.

•While the reported TTI prevalence (25.7%) is contextualized through comparison with prior studies, the inclusion of Saudi Arabia as a comparator (Ref. 16) in the discussion is epidemiologically inappropriate and potentially misleading. Saudi Arabia represents a distinct donor population with markedly different baseline infection prevalence, donor recruitment strategies, and screening protocols compared with Sub-Saharan African settings. Comparative interpretations should therefore be limited to regions with broadly comparable epidemiological and healthcare contexts.

•Some comments about figures and tables here and there!

oIn Figure 1:

The flow diagram oversimplifies cohort selection and does not adequately describe exclusion steps or eligibility criteria, limiting reproducibility and interpretability.

The figure lacks a descriptive title and legend, and visual elements appear decorative rather than informative.

oIn Table 1:

The high proportion of undocumented variables severely limits interpretability and undermines the validity of comparisons.

Testing both continuous and categorized age variables introduces redundancy and inflates the number of statistical comparisons.

Blood group frequencies differ substantially across populations and may be confounded by ethnicity, geography, or donor selection. It is highly recommended to clearly state that this table is univariate.

Several comparisons involve extremely small cell counts, violating assumptions of chi-square testing.

•In summary, while the study addresses an important question and is based on a valuable dataset, substantial revisions are needed to improve conceptual coherence, epidemiological interpretation, methodological transparency, and biological depth. Addressing the points above would significantly enhance the rigor, clarity, and overall impact of the manuscript.

Reviewer #4: Dear Authors

Thank you for submitting your research for publication. The manuscript addresses an important topic. However, several methodological, conceptual, and reporting issues must be addressed before the manuscript can be considered for publication. Major revisions are required to ensure scientific rigor, clarity, and alignment between study aims, analysis, and conclusions.

Major revision issues:

1. There is a mismatch between stated study aims and analyses conducted. The stated study aim includes addressing the reasons behind high blood discard rates beyond TTIs; however, the analyses presented focus exclusively on transfusion-transmissible infections. Non-TTI causes of discard (e.g., expiry, inadequate volume, storage or handling issues) are not investigated.

** Recommendation: Revise the study aim to align with the analyses conducted or expand the analysis to include non-TTI discard reasons. Adjust conclusions accordingly.

2. The conclusion that individuals with secondary education have a higher risk of infection compared to those with “not documented education level” is not methodologically justified. The “not documented” category does not represent a valid comparison group and raises concerns regarding misclassification and missing data bias. Authors should justify the inclusion of this category as a reference group or reanalyse the data by treating undocumented education as missing. Comparative analyses should be restricted to clearly defined education levels. The authors should refrain from making risk-based inferences using this category and revise both the analysis and conclusions.

** Recommendation: Remove or reanalyse education-level comparisons, treating undocumented education as missing data. Condense or remove the related discussion and reframe it as a limitation.

3. Limited screening infrastructure would be expected to result in underestimation, not higher observed prevalence as stated in lines 68-71.

** Recommendation: Clarify the mechanism or revise wording to avoid causal inconsistency between detection capacity and observed prevalence.

Detailed comments/recommendations:

Abstract:

Line 46: “further mechanistic investigation...”

** Rephrase or explain “mechanistic investigation”

Introduction:

Line 50-51: “Blood transfusion remains a critical... that is essential for treating various conditions including surgical procedures, trauma management…”.

** This statement is imprecise. Blood transfusion is a supportive, life-saving intervention used when clinically indicated (e.g., perioperative haemorrhage or trauma-related blood loss), rather than a treatment for surgery itself. The authors are encouraged to revise this sentence for clinical accuracy and clarity.

Line 68-71: “These rates are substantially higher than those observed in high-income countries, reflecting the complex interplay of factors including high community prevalence of these infections, reliance on replacement and first-time donors, and limitations in screening infrastructure”.

** The attribution of higher observed prevalence to “limitations in screening infrastructure” is conceptually unclear. Limited screening capacity would be expected to result in underestimation rather than higher measured prevalence. This contradiction should be addressed or the wording revised.

Line 83-85: “Furthermore, the economic impact of discarded units, combined with the costs of screening and confirmatory testing, places additional strain on already resource-constrained blood transfusion service”.

** This statement, although might be correct, does not directly relate to the study aim or contribute to the interpretation of the results. Its inclusion appears tangential and should be justified or omitted.

Line 88-89: “This study aims to address the reasons behind the high discard rates and explore the factors contributing to the prevalence of TTIs in the province”.

** The analyses presented focus exclusively on transfusion-transmissible infections. Non-TTI causes of discard are not investigated. The authors should either revise the study aim to align with the analyses conducted or expand the analysis to include non-TTI discard reasons. Adjust conclusions accordingly.

Result:

Line 149: “A total of 2,667 participants were included in the analysis, of whom 686 (25.7%) had screened positive for infected blood.”

** Rephrase statement for clarity and accuracy.

Line 153-154: “Participants with a secondary education had a significantly higher infection prevalence (31.5%) compared to those whose education level was not documented (25.0%; p=0.012).”.

** As mentioned above, authors should justify the inclusion of “not documented education level” category as a reference group or reanalyse the data by treating undocumented education as missing. Comparative analyses should be restricted to clearly defined education levels.

Discussion:

Line 210-222:

** The extended discussion of education level relies on a comparison with the “not documented” category, which the authors themselves acknowledge represents missing and heterogeneous data. As such, inferential comparisons and adjusted analyses using this category are not valid. The discussion remains speculative and introduces unsupported claims regarding the risk profile of individuals with missing education data. I recommend removing this paragraph and instead briefly noting the issue of missing educational data as a limitation.

Limitations:

** The limitations should be expanded on to include a more discussion of the potential confounders.

7. PLOS authors have the option to publish the peer review history of their article (what does this mean?). If published, this will include your full peer review and any attached files.). If published, this will include your full peer review and any attached files.). If published, this will include your full peer review and any attached files.). If published, this will include your full peer review and any attached files.

...

Reviewer #1: No

Reviewer #3: No

Reviewer #4: No

---

## [Author Response · Author response to Decision Letter 2]

20 Jan 2026

Point by Point Response to reviewer comments

Review Comments to the Author

Reviewer #1: The author to some extent has responded to the comments, It's a decent effort , though it lacks depth. Just here one comment, in tables 1, 2, age group 14-24 years. Does 14 years eligible for blood donation in Zambia?!

Usually Standard age of donation is 18 to 65 years old, as organizations focus on donors who are in good health, meet weight standards, hemoglobin standards, and have a stable medical history to ensure donor safety and blood quality.

The minimum age is usually 16 or 17 with parental consent in some places. But at 14 it is restricted as individuals are still developing, and to protect their health.

Response

We thank the reviewer for the insight comment and feedback. We agree with the reviewer’s comment. The minimum age permitted to donate blood in Zambia and in our study was 16 years of age. We sincerely apologies as this was a typographic error. We have revised tables 1 and 2, age group from 14-24 years to 16-24 years.

Reviewer #3: This cross sectional study present the burden and correlates of bloodborne pathogens among blood donors in a Zambian cohort, particular focus on identifying demographic, behavioral, and operational factors associated with transfusion-transmitted infections (TTI) prevalence. However, my concerns/suggestions are not necessarily a reflection of the quality of the work, as follows:

•Overall, the introduction would benefit from a clearer and more cohesive rationale, with smoother transitions between paragraphs. In particular, the discussion of cancer treatment implications lacks a sufficiently developed biological or clinical rationale. Strengthening the mechanistic or pathophysiological basis for this link would help justify the study’s relevance and potential impact.

Response

We thank the reviewer on this insight comment. We have revised the paragraphs to improve clinical rationality and to make logical flow. We have also removed cancer treatment from the introduction.

•While the reported TTI prevalence (25.7%) is contextualized through comparison with prior studies, the inclusion of Saudi Arabia as a comparator (Ref. 16) in the discussion is epidemiologically inappropriate and potentially misleading. Saudi Arabia represents a distinct donor population with markedly different baseline infection prevalence, donor recruitment strategies, and screening protocols compared with Sub-Saharan African settings. Comparative interpretations should therefore be limited to regions with broadly comparable epidemiological and healthcare contexts.

Response

We thank the reviewer on this insight comment. We have revised and removed the Ref. 16

•Some comments about figures and tables here and there!

oIn Figure 1:

The flow diagram oversimplifies cohort selection and does not adequately describe exclusion steps or eligibility criteria, limiting reproducibility and interpretability.

The figure lacks a descriptive title and legend, and visual elements appear decorative rather than informative.

Response

We thank the reviewer on this insight comment. We have revised and removed the figure.

oIn Table 1:

The high proportion of undocumented variables severely limits interpretability and undermines the validity of comparisons.

Response

We thank the reviewer on this insight comment. We have revised and removed undocumented category. We have mentioned this as a limitation.

Testing both continuous and categorized age variables introduces redundancy and inflates the number of statistical comparisons.

Response

We thank the reviewer on this insight comment. We have revised and removed continuous variable for age.

Blood group frequencies differ substantially across populations and may be confounded by ethnicity, geography, or donor selection. It is highly recommended to clearly state that this table is univariate.

Response

We thank the reviewer on this insight comment. We have revised and replace bivariate to univariate.

Several comparisons involve extremely small cell counts, violating assumptions of chi-square testing.

Response

We thank the reviewer on this insight comment. We have added fishers exact test for small cel counts. The statement now reads “Chi-square tests or fishers exact test for small cel counts was used for categorical variables”.

•In summary, while the study addresses an important question and is based on a valuable dataset, substantial revisions are needed to improve conceptual coherence, epidemiological interpretation, methodological transparency, and biological depth. Addressing the points above would significantly enhance the rigor, clarity, and overall impact of the manuscript.

Response

We thank the reviewer on this insight comment. We have revised the entire manuscript attending to the reviewers’ comments.

Reviewer #4: Dear Authors

Thank you for submitting your research for publication. The manuscript addresses an important topic. However, several methodological, conceptual, and reporting issues must be addressed before the manuscript can be considered for publication. Major revisions are required to ensure scientific rigor, clarity, and alignment between study aims, analysis, and conclusions.

Major revision issues:

1. There is a mismatch between stated study aims and analyses conducted. The stated study aim includes addressing the reasons behind high blood discard rates beyond TTIs; however, the analyses presented focus exclusively on transfusion-transmissible infections. Non-TTI causes of discard (e.g., expiry, inadequate volume, storage or handling issues) are not investigated.

** Recommendation: Revise the study aim to align with the analyses conducted or expand the analysis to include non-TTI discard reasons. Adjust conclusions accordingly.

Response

We thank the reviewer for the insightful comment provided. We have revised the aim of the study to suite the investigation conducted. The aim of the study now reads “This study aims to determine the burden and correlates of transfusion-transmissible infections among blood donors in a Zambian cohort, with particular focus on identifying demographic and donor-related factors associated with TTI prevalence. By elucidating these factors, the study seeks to inform strategies for improving blood safety and reducing TTI-related discards in the region.”

2. The conclusion that individuals with secondary education have a higher risk of infection compared to those with “not documented education level” is not methodologically justified. The “not documented” category does not represent a valid comparison group and raises concerns regarding misclassification and missing data bias. Authors should justify the inclusion of this category as a reference group or reanalyse the data by treating undocumented education as missing. Comparative analyses should be restricted to clearly defined education levels. The authors should refrain from making risk-based inferences using this category and revise both the analysis and conclusions.

** Recommendation: Remove or reanalyse education-level comparisons, treating undocumented education as missing data. Condense or remove the related discussion and reframe it as a limitation.

Response

We thank the reviewer on this insight comment. We have revised the manuscript by treating education undocumented education as missing. We have further removed this from the discussion section and have added it as a limitation.

3. Limited screening infrastructure would be expected to result in underestimation, not higher observed prevalence as stated in lines 68-71.

** Recommendation: Clarify the mechanism or revise wording to avoid causal inconsistency between detection capacity and observed prevalence.

Response

We thank the reviewer on this insight comment. We have revised the manuscript to avoid a causal inconsistency.

Detailed comments/recommendations:

Abstract:

Line 46: “further mechanistic investigation...”

** Rephrase or explain “mechanistic investigation”

Response

We thank the reviewer on this insight comment. We have revised the phrase and replaced it with “suggesting a potential biological basis that requires further study”.

Introduction:

Line 50-51: “Blood transfusion remains a critical... that is essential for treating various conditions including surgical procedures, trauma management…”.

** This statement is imprecise. Blood transfusion is a supportive, life-saving intervention used when clinically indicated (e.g., perioperative haemorrhage or trauma-related blood loss), rather than a treatment for surgery itself. The authors are encouraged to revise this sentence for clinical accuracy and clarity.

Response

We thank the reviewer on this insight comment. We have revised the phrase for clinical accuracy and clarity.

Line 68-71: “These rates are substantially higher than those observed in high-income countries, reflecting the complex interplay of factors including high community prevalence of these infections, reliance on replacement and first-time donors, and limitations in screening infrastructure”.

** The attribution of higher observed prevalence to “limitations in screening infrastructure” is conceptually unclear. Limited screening capacity would be expected to result in underestimation rather than higher measured prevalence. This contradiction should be addressed or the wording revised.

Response

We thank the reviewer on this insight comment. We have revised the phrase to resolve the contradiction in the wording.

Line 83-85: “Furthermore, the economic impact of discarded units, combined with the costs of screening and confirmatory testing, places additional strain on already resource-constrained blood transfusion service”.

** This statement, although might be correct, does not directly relate to the study aim or contribute to the interpretation of the results. Its inclusion appears tangential and should be justified or omitted.

Response

We thank the reviewer on this insight comment. We have revised and omitted the phrase.

Line 88-89: “This study aims to address the reasons behind the high discard rates and explore the factors contributing to the prevalence of TTIs in the province”.

** The analyses presented focus exclusively on transfusion-transmissible infections. Non-TTI causes of discard are not investigated. The authors should either revise the study aim to align with the analyses conducted or expand the analysis to include non-TTI discard reasons. Adjust conclusions accordingly.

Response

We thank the reviewer on this insight comment. We have revised the study aim to align with the analyses conducted.

Result:

Line 149: “A total of 2,667 participants were included in the analysis, of whom 686 (25.7%) had screened positive for infected blood.”

** Rephrase statement for clarity and accuracy.

Response

We thank the reviewer on this insight comment. We have revised the statement for clarity and accuracy.

Line 153-154: “Participants with a secondary education had a significantly higher infection prevalence (31.5%) compared to those whose education level was not documented (25.0%; p=0.012).”.

** As mentioned above, authors should justify the inclusion of “not documented education level” category as a reference group or reanalyse the data by treating undocumented education as missing. Comparative analyses should be restricted to clearly defined education levels.

Response

We thank the reviewer on this insight comment. We have excluded undocumented cases from comparative analyses. We have also added footnote for clarity.

Discussion:

Line 210-222:

** The extended discussion of education level relies on a comparison with the “not documented” category, which the authors themselves acknowledge represents missing and heterogeneous data. As such, inferential comparisons and adjusted analyses using this category are not valid. The discussion remains speculative and introduces unsupported claims regarding the risk profile of individuals with missing education data. I recommend removing this paragraph and instead briefly noting the issue of missing educational data as a limitation.

Response

We thank the reviewer on this insight comment. We have removed this paragraph from

The discussion and instead have replace with missing education data as a limitation

Limitations:

** The limitations should be expanded on to include a more discussion of the potential confounders.

Response

We thank the reviewer on this insight comment. We have expanded on limitations

---

## [Decision Letter · Decision Letter 2]

3 Feb 2026

PONE-D-25-55464R2The Burden and Correlates of Bloodborne Pathogens in a Zambian Blood Donor CohortPLOS One

Dear Dr. Chisompola,

Thank you for submitting your manuscript to PLOS ONE. After careful consideration, we feel that it has merit but does not fully meet PLOS ONE’s publication criteria as it currently stands. Therefore, we invite you to submit a revised version of the manuscript that addresses the points raised during the review process.

We look forward to receiving your revised manuscript.

Kind regards,

Nneoma Confidence JeanStephanie Anyanwu, Ph.D.

Academic Editor

PLOS One

**Journal Requirements:**

Reviewers' comments:

Reviewer's Responses to Questions

Comments to the Author

1. If the authors have adequately addressed your comments raised in a previous round of review and you feel that this manuscript is now acceptable for publication, you may indicate that here to bypass the “Comments to the Author” section, enter your conflict of interest statement in the “Confidential to Editor” section, and submit your "Accept" recommendation.

Reviewer #3: (No Response)

Reviewer #4: (No Response)

2. Is the manuscript technically sound, and do the data support the conclusions?

Reviewer #3: No

Reviewer #4: Partly

3. Has the statistical analysis been performed appropriately and rigorously? 

Reviewer #3: No

Reviewer #4: No

4. Have the authors made all data underlying the findings in their manuscript fully available?

The PLOS Data policy requires authors to make all data underlying the findings described in their manuscript fully available without restriction, with rare exception (please refer to the Data Availability Statement in the manuscript PDF file). The data should be provided as part of the manuscript or its supporting information, or deposited to a public repository. For example, in addition to summary statistics, the data points behind means, medians and variance measures should be available. If there are restrictions on publicly sharing data—e.g. participant privacy or use of data from a third party—those must be specified.requires authors to make all data underlying the findings described in their manuscript fully available without restriction, with rare exception (please refer to the Data Availability Statement in the manuscript PDF file). The data should be provided as part of the manuscript or its supporting information, or deposited to a public repository. For example, in addition to summary statistics, the data points behind means, medians and variance measures should be available. If there are restrictions on publicly sharing data—e.g. participant privacy or use of data from a third party—those must be specified.requires authors to make all data underlying the findings described in their manuscript fully available without restriction, with rare exception (please refer to the Data Availability Statement in the manuscript PDF file). The data should be provided as part of the manuscript or its supporting information, or deposited to a public repository. For example, in addition to summary statistics, the data points behind means, medians and variance measures should be available. If there are restrictions on publicly sharing data—e.g. participant privacy or use of data from a third party—those must be specified.requires authors to make all data underlying the findings described in their manuscript fully available without restriction, with rare exception (please refer to the Data Availability Statement in the manuscript PDF file). The data should be provided as part of the manuscript or its supporting information, or deposited to a public repository. For example, in addition to summary statistics, the data points behind means, medians and variance measures should be available. If there are restrictions on publicly sharing data—e.g. participant privacy or use of data from a third party—those must be specified.

Reviewer #3: No

Reviewer #4: Yes

5. Is the manuscript presented in an intelligible fashion and written in standard English?

Reviewer #3: Yes

Reviewer #4: Yes

6. Review Comments to the Author

Reviewer #3: Thank you for the revised manuscript. The revision shows clear improvement and addresses several earlier concerns; however, some important issues remain. In particular, the handling of the “undocumented” education variable is inconsistent, as it continues to appear in adjusted analyses and interpretations even though it is described as excluded. In addition, there are residual internal inconsistencies, including an incorrect age category and incomplete alignment between the Methods, Tables, Results, and Abstract. Addressing these points in a focused revision will be necessary to ensure methodological clarity and internal consistency.

Reviewer #4: Dear Authors,

We all agree that "Undocumented education" is not a true educational category.

The inclusion of “undocumented education level” as a predictor in univariable analysis remains methodologically inappropriate. This category represents missing data rather than a true exposure, and odds ratios derived from it are not interpretable, regardless of statistical significance. The reported association likely reflects patterns in data completeness rather than a meaningful relationship with infection risk. I strongly recommend removing this variable from inferential analyses and treating missing education as a limitation rather than a risk factor.

There is a discrepancy between the abstract and the Results section. The authors should reconcile these differences, clarify whether the reported estimate is from univariable or multivariable analysis, and ensure that the abstract accurately reflects the final results presented in the manuscript.

In addition, given the methodological concerns regarding the interpretability of the “undocumented education level” category, the authors should carefully reconsider whether this variable should be highlighted in the abstract at all.

Best wishes

7. PLOS authors have the option to publish the peer review history of their article (what does this mean?). If published, this will include your full peer review and any attached files.). If published, this will include your full peer review and any attached files.). If published, this will include your full peer review and any attached files.). If published, this will include your full peer review and any attached files.

Do you want your identity to be public for this peer review? For information about this choice, including consent withdrawal, please see our Privacy Policy....

Reviewer #3: No

Reviewer #4: No

---

## [Author Response · Author response to Decision Letter 3]

4 Feb 2026

Point by Point responses to Reviewers’ comments

We thank the reviewers for their helpful comments. The manuscript has been revised to address their feedback, as outlined in our responses.

Reviewer #3: Thank you for the revised manuscript. The revision shows clear improvement and addresses several earlier concerns; however, some important issues remain. In particular, the handling of the “undocumented” education variable is inconsistent, as it continues to appear in adjusted analyses and interpretations even though it is described as excluded. In addition, there are residual internal inconsistencies, including an incorrect age category and incomplete alignment between the Methods, Tables, Results, and Abstract. Addressing these points in a focused revision will be necessary to ensure methodological clarity and internal consistency.

Response

We thank the reviewers for their insightful feedback and observations. We have removed ‘undocumented education’ from all inferential analyses and clarified that it represents missing data. All age categories are now consistent, and the Abstract has been aligned with the final Results. We thank the reviewers for their insightful comments, which have strengthened the manuscript.

Reviewer #4: Dear Authors,

We all agree that "Undocumented education" is not a true educational category.

The inclusion of “undocumented education level” as a predictor in univariable analysis remains methodologically inappropriate. This category represents missing data rather than a true exposure, and odds ratios derived from it are not interpretable, regardless of statistical significance. The reported association likely reflects patterns in data completeness rather than a meaningful relationship with infection risk. I strongly recommend removing this variable from inferential analyses and treating missing education as a limitation rather than a risk factor.

Response

We thank the reviewers for their insightful feedback and observations. We have revised and reviewed the manuscript. We have corrected as suggested.

There is a discrepancy between the abstract and the Results section. The authors should reconcile these differences, clarify whether the reported estimate is from univariable or multivariable analysis, and ensure that the abstract accurately reflects the final results presented in the manuscript.

Response

We thank the reviewers for their insightful feedback and observations. We have revised and have clearly indicated that the multivariable analysis was conducted.

In addition, given the methodological concerns regarding the interpretability of the “undocumented education level” category, the authors should carefully reconsider whether this variable should be highlighted in the abstract at all.

Response

We thank the reviewers for their insightful feedback. We have revised the abstract and have ensured that education status is not appearing. We have further omitted education status in multivariable analysis. But we have shown education under descriptive as percentages in table 1 participant characteristics. We have also updated the method section to add a statement of no inclusion of education status.

---

## [Decision Letter · Decision Letter 3]

24 Mar 2026

The Burden and Correlates of Bloodborne Pathogens in a Zambian Blood Donor Cohort

PONE-D-25-55464R3

Dear Dr. Chisompola,

We’re pleased to inform you that your manuscript has been judged scientifically suitable for publication and will be formally accepted for publication once it meets all outstanding technical requirements.

Kind regards,

Nneoma Confidence JeanStephanie Anyanwu, Ph.D.

Academic Editor

PLOS One

Additional Editor Comments (optional):

Reviewers' comments:

Reviewer's Responses to Questions

**Comments to the Author**

**Reviewer #3: (No Response)**

2. Is the manuscript technically sound, and do the data support the conclusions?

**Reviewer #3: Partly**

**3. Has the statistical analysis been performed appropriately and rigorously? **

**Reviewer #3: Yes**

4. Have the authors made all data underlying the findings in their manuscript fully available?

The PLOS Data policy requires authors to make all data underlying the findings described in their manuscript fully available without restriction, with rare exception (please refer to the Data Availability Statement in the manuscript PDF file). The data should be provided as part of the manuscript or its supporting information, or deposited to a public repository. For example, in addition to summary statistics, the data points behind means, medians and variance measures should be available. If there are restrictions on publicly sharing data—e.g. participant privacy or use of data from a third party—those must be specified.requires authors to make all data underlying the findings described in their manuscript fully available without restriction, with rare exception (please refer to the Data Availability Statement in the manuscript PDF file). The data should be provided as part of the manuscript or its supporting information, or deposited to a public repository. For example, in addition to summary statistics, the data points behind means, medians and variance measures should be available. If there are restrictions on publicly sharing data—e.g. participant privacy or use of data from a third party—those must be specified.requires authors to make all data underlying the findings described in their manuscript fully available without restriction, with rare exception (please refer to the Data Availability Statement in the manuscript PDF file). The data should be provided as part of the manuscript or its supporting information, or deposited to a public repository. For example, in addition to summary statistics, the data points behind means, medians and variance measures should be available. If there are restrictions on publicly sharing data—e.g. participant privacy or use of data from a third party—those must be specified.requires authors to make all data underlying the findings described in their manuscript fully available without restriction, with rare exception (please refer to the Data Availability Statement in the manuscript PDF file). The data should be provided as part of the manuscript or its supporting information, or deposited to a public repository. For example, in addition to summary statistics, the data points behind means, medians and variance measures should be available. If there are restrictions on publicly sharing data—e.g. participant privacy or use of data from a third party—those must be specified.

**Reviewer #3: Yes**

5. Is the manuscript presented in an intelligible fashion and written in standard English?

**Reviewer #3: No**

6. Review Comments to the Author

**Reviewer #3: (No Response)**

7. PLOS authors have the option to publish the peer review history of their article (what does this mean?). If published, this will include your full peer review and any attached files.). If published, this will include your full peer review and any attached files.). If published, this will include your full peer review and any attached files.). If published, this will include your full peer review and any attached files.

...

**Reviewer #3: No**

---

## [Editor Report · Acceptance letter]

PONE-D-25-55464R3

PLOS One

Dear Dr. Chisompola,

I'm pleased to inform you that your manuscript has been deemed suitable for publication in PLOS One. Congratulations! Your manuscript is now being handed over to our production team.

Kind regards,

on behalf of

Dr. Nneoma Confidence JeanStephanie Anyanwu

Academic Editor

PLOS One